# Can ICT investment promote green development? new insights from highly polluting listed enterprises in China

**Yingdong Wang, Qingbo Li**●*

School of Finance, Zhongnan University of Economics and Law, Wuhan, China

* qingboli6688@163.com

**Data Availability Statement:** All relevant data are within the paper and its Supporting Information files.

**Funding:** The author(s) received no specific funding for this work.

## Abstract

Green development is related to the long-term sustainable development of enterprises, but there is little literature on the impact of information and communication technology (ICT) investment on green development of enterprises. To fill this gap, based on the sample of A-share high-polluting industry enterprises in China's Shanghai and Shenzhen stock markets from 2007 to 2019, this paper uses the dynamic panel threshold model to deeply study the non-linear impact of ICT investment on green development. This paper found that: (1) there is a threshold effect on the impact of ICT investment on green development of enterprises in high-polluting industries, and when ICT investment breaks through the threshold value, it can promote green development. This conclusion remains robust after the estimation of Multi-period DID model and instrumental variable; (2) ICT investment mainly promotes green development of enterprises by improving production efficiency and promoting green technology innovation; (3) ICT investment has a significant impact on green development of enterprises with higher management level, heavy industrial enterprises and non-state-owned enterprises, but has no significant impact on green development of enterprises with lower management level, light industrial enterprises and state-owned enterprises. And digital technology investment has a stronger role in promoting green development than traditional ICT investment. This paper proposes that enterprises in high-pollution industries can increase ICT investment to promote green development.

## 1. Introduction

Green development is a way of development that decouples economic growth from resource consumption, carbon emissions and environmental damage [1], and in essence, it is to achieve the synergy of "green" and "development". Green development of enterprises is not only related to the response to global climate change and environmental crisis, but also to the future market competitiveness, social responsibility and long-term sustainable development of enterprises [2,3]. At present, the global resource consumption rate is still higher than the natural regeneration rate of the earth. According to the "CO2 emissions 2023" released by the International Energy Agency (IEA), the global energy-related carbon dioxide emissions in 2023 were

**Competing interests:** The authors have declared that no competing interests exist.

37.4 billion tons and reached a record high again, which exacerbated the risk of climate change. Major countries in the world are actively promoting green development strategies and China announced in 2017 that it would achieve carbon neutrality around 2060. The European Union proposed the "European Green New Deal" and introduced measures to achieve carbon neutrality in 2019. And the Biden administration announced its re-entry into the Paris Agreement in 2021. Green products and services are highly valued by consumers and companies with good environmental credentials will enjoy an increasing competitive advantage [4,5]. In summary, achieving green development is a necessary and urgent choice for enterprises.

Enterprises in high-polluting industries are the main responsible enterprises for pollution emissions and they will implement green transformation more purposefully. Exploring how to achieve their green development is also crucial for achieving the country's overall green development. Enterprises in high-pollution industries are facing various challenges such as environmental pressure, regulatory pressure, social responsibility and public opinion. They must formulate practical strategies to achieve green development. Highly polluting industries such as coal and chemical industries face strict emission standards and must invest in technological transformation to reduce pollutant emissions. Governments continue to strengthen environmental protection regulations, and enterprises in high-pollution industries need to increase investment in equipment upgrades, employee training, and environmental monitoring, which increases the cost of corporate compliance. In addition, the public's expectations for corporate environmental responsibility have increased, and companies in high-pollution industries need to improve environmental performance and focus on social responsibility.

However, it is not easy for enterprises in high-pollution industries to finally achieve green development and they mainly face the following three difficulties. Firstly, companies adopting green technologies and clean energy require higher investment costs and face longer payback periods [6], especially when the company's technology maturity is not high. Secondly, the efficiency of cooperation between enterprises and its supply chain upstream and downstream enterprises is low. Green development requires the participation and support of enterprises in the entire supply chain and involves temporary changes and coordination of cooperative enterprises. Thirdly, in the face of greater market competition pressure, companies are less likely to gain a competitive advantage by raising environmental standards, but more likely to reduce product prices or improve product quality.

Information and Communications Technology (ICT) includes both traditional ICT, such as network technology and communication technology, and a new generation of ICT, that is, digital technology, such as big data, blockchain, cloud computing and artificial intelligence. In this paper, ICT investment refers to the investment of enterprises in computer hardware, communication equipment and various software. ICT investment has outstanding advantages in solving the problem of green development of enterprises in high-pollution industries. First, ICT investment can help enterprises to achieve intelligent monitoring and management of production processes, identify potential energy-saving emission reduction opportunities and optimization schemes, and thus reduce pollutant emissions. According to Moore's law, the function of ICT products will be stronger and stronger, and the price will be lower and lower. Therefore, the cost of ICT investment will be lower than that of large-scale purchase of green technology and replacement of production equipment. Second, ICT-based digital supply chain management system can realize information sharing and collaboration among enterprises in the supply chain [7]. Enterprises can use the supply chain management system to better cooperate with stakeholders, improve the efficiency of cooperation, and achieve the purpose of pollutant emission reduction. At the same time, enterprises can also better carry out green collaborative innovation with stakeholders. Third, through the ICT-based network platform, enterprises can learn a wealth of green technology knowledge from the outside world. Through

the collaborative innovation platform, enterprises can actively utilize external resources for green innovation, i.e., research and development cooperation with domestic and foreign universities, scientific research institutions and enterprises with complementary resources to their own. These are conducive to enhancing the green technology innovation ability and market competitiveness of enterprises [8].

The existing literature mainly studies the impact and impact mechanism of ICT investment on green development at the macro and meso levels, but there is little literature on the relationship between the two at the micro level. At the macro and meso levels, ICT investment can promote green development of China's provinces by promoting technological innovation, reducing the distortion of industrial structure, improving the efficiency of resource allocation and enhancing the level of human capital [9]. And the ICT-based Industry 4.0 model has significantly improved the efficiency of energy use in the industrial sector and played a role in environmental protection [10]. At the micro level, Ye et al. named ICT investment in 2016 and beyond as digital technology investment and studied the linear impact of digital technology investment on the environmental performance of enterprises in China's high pollution industries [11]. However, digital technology investment only belongs to China's new stage of ICT investment, and traditional ICT investment such as Internet and telephone is also important and of research interest.

Considering the possible impact of ICT investment on the green development of enterprises in reality and the research status of existing literature, this paper intends to explore: Can ICT investment promote green development of enterprises in high pollution industries? Is this effect nonlinear? What are the mechanisms by which ICT investments influence green development of enterprises?

The marginal contribution of this paper lies in, firstly, the existing literature mostly studies the impact of ICT investment on green development from the macro and meso levels, while this paper reveals the impact of ICT investment on green development from the enterprise level, which is a useful supplement to the study of the relationship between the two at the macro and meso levels. Specifically, on the one hand, the research in this paper expands the analysis of the influencing factors of green development of enterprises. Compared with the influence of environmental regulation that scholars have paid attention to in the past, this paper focuses on the role of ICT, which has an increasing impact on the economy and society. On the other hand, the research in this paper also expands the analysis of the impact of ICT investment. ICT investment can not only affect economic growth and industrial structure transformation, but also promote green technology innovation and green transformation of enterprises.

Secondly, this paper deeply studies the non-linear impact of ICT investment on the green development of enterprises, which also expands the research on the linear relationship between the two. Although ICT products have a great impact on production efficiency and technological innovation, they are also high-energy-consuming products and have a negative impact on the environment. ICT products generally achieve the effect of increasing returns to scale through the network effect. Combined with the threshold characteristics of network effect and the negative impact of ICT products on the environment, ICT investment may have a threshold effect on the green development of enterprises. When ICT investment exceeds a certain level, it can promote the green development of enterprises, which is not considered in depth in previous studies.

Thirdly, the research conclusions of this paper provide new insights for the application of ICT products in the green development of enterprises. On the one hand, the green transformation of enterprises does not necessarily require a large amount of money to replace production equipment or purchase green technology, but can achieve networked and digital

transformation of production and management, save resources and reduce pollution emissions and improve their own green technology innovation level through ICT investment. On the other hand, when enterprises carry out ICT investment and digital transformation, in addition to applying ICT to customer relationship management, they can also apply ICT to intelligent manufacturing, data analysis and decision support and the construction of collaborative innovation platform, and use ICT to promote the green development.

The paper proceeds as follows. Section 2 is literature review. Section 3 is theoretical analysis and research hypotheses. Section 4 is research design. Section 5 is analysis of empirical results. Section 6 is conclusion and policy recommendations.

## 2. Literature review

### 2.1 Research on the impact effect of ICT investment

ICT investment can mainly affect the upgrading of industrial structure, production efficiency and technological innovation. First, ICT investment can promote the upgrading of industrial structure. Based on the panel data of Chinese provinces, Yan et al. found that ICT investment can promote the upgrading of industrial structure through the path of capital deepening [12]. Second, ICT investment can improve production efficiency. Based on the panel data of "The Belt and Road" countries, Shen et al. found that increasing the market size of e-commerce can improve the efficiency of mineral resources use [13]. ICT investment can improve the total factor productivity (TFP) of Chinese enterprises by reducing information search costs, improving information processing capabilities and reducing internal capital misallocation [14]. ICT investment has also enhanced the ability of small and medium-sized enterprises in India to operate smoothly, improved their organizational capabilities and performance, promoted communication and coordination between enterprises and partners, and ultimately improved their TFP [15]. The impact of ICT investment on production efficiency also has a spillover effect, and the network externality of e-commerce capital stock has a positive spillover effect on the production efficiency of enterprises in other industries [16]. Third, ICT investment can promote technological innovation. ICT investment amplifies the cultural influence related to the pursuit of goals and is beneficial to technological innovation [17]. Cuevas-Vargas et al. found that ICT investment promotes open innovation by improving knowledge absorptive capacity based on the data of ICT investment obtained from the questionnaire survey [18]. Online communication platforms will also promote cooperative innovation among members on the platform [19].

### 2.2 Research on the influencing factors of green development

There are four main factors affecting green development, which are institutional arrangements, resource endowments, production efficiency and green technology innovation. First, institutional arrangements can significantly affect green development. The two main ways of government environmental regulation are sewage charges and environmental subsidies, which can maximize green development of enterprises when they are implemented in combination [20]. The carbon emissions trading system can realize cost transfer between enterprises with different energy efficiency through the market itself and can improve the green total factor productivity of enterprises [21]. Second, resource endowments will affect green development. Zhang et al. found that resource endowment can significantly promote green development of mature cities and regenerative cities with high resource utilization efficiency, but it will hinder green development of declining cities and growing cities with low resource utilization efficiency [22]. Third, production efficiency is conducive to promoting green development. Improving manufacturing efficiency in the industrial sector contributes to reduce industrial

pollution and achieve cleaner production [23]. TFP improvement is good for reducing the redundancy of production input and can promote green development of China's resource-based enterprises [24]. Fourth, green technology innovation is beneficial to promoting green development. Green technology innovation has significantly improved the carbon emission efficiency of developed countries with the goal of carbon neutrality [25], and green technology innovation can help Chinese industrial enterprises improve their environmental performance score [26].

## 2.3 Research on the relationship between ICT investment and green development

Most scholars have studied the impact of ICT investment on green development at the macro and meso levels, but few scholars have studied the relationship between the two at the enterprise level. At the macro level, Bhujabal et al. found that ICT investment can reduce the degree of environmental pollution in Asia-Pacific countries [27]. ICT investment mainly promotes green development of Chinese provinces by strengthening technological innovation, correcting industrial structure distortion, improving resource allocation efficiency and human capital level [9]. But Jakada et al. found that major African economies will increase carbon dioxide emissions after ICT investment [28]. And the possible explanation is that the low level of ICT management in these countries has led to inefficient use of ICT products. At the meso level, Wachnik et al. found that the ICT-based Industry 4.0 model significantly improved the efficiency of energy use in the industrial sector and reduced its total energy use [10]. The pilot policy of "Broadband China" has improved the eco-efficiency of Chinese cities by strengthening green technology innovation, promoting industrial structure upgrading and improving resource allocation efficiency [29]. At the micro level, Ye et al. studied the impact of digital technology investment on the environmental performance of Chinese enterprises and found that digital technology investment can improve the environmental performance of enterprises [11]. However, they did not study the impact of all types of ICT investment from a longer period of time, nor did it examine the nonlinear relationship between ICT investment and corporate environmental performance.

## 3. Theoretical analysis and research hypothesis

### 3.1 The nonlinear impact of ICT investment on green development of enterprises in highly polluting industries

ICT, as a general purpose technology, makes ICT products highly permeable and can easily affect various economic activities such as production and consumption. The network effect theory holds that the value of the network is proportional to the number of nodes in the network [30]. For example, the production department of the enterprise can use sensor and network technology to connect various equipment, so that each production equipment of the enterprise can form an internal network, so as to realize data exchange and real-time monitoring between equipment, which is conducive to improving the utilization rate of equipment and reducing production costs. By constructing a collaborative innovation platform, enterprises can also form an external innovation network with universities, scientific research institutions or stakeholders to enhance their knowledge absorption capacity and resource integration ability, so as to improve the level of green technology innovation. The network formed by ICT investment in enterprises mainly includes the network between production equipment, the network between internal departments and the network between enterprises and external related subjects. With the gradual use of computers and Internet services by

many enterprises or various departments within the enterprise, the scale effect of the network will gradually become prominent.

Specifically, first, ICT investment can help enterprises achieve networked and digital transformation, and enterprises will build production equipment network, internal department network and online sales network, which can improve their production efficiency and promote green development [14]. Networked and digital production such as "Smart Factory" can help enterprises realize the network and intelligence of production lines. Through automation technology, robots and intelligent control systems, enterprises can improve the efficiency of resource use and reduce the marginal cost of production. At the same time, the production equipment network can also better monitor and analyze the energy consumption in the production process and find opportunities for energy saving and consumption reduction. Network and digital communication platform can improve the efficiency of communication and cooperation between enterprise personnel [31], such as enterprise online work platform or online conference software. This allows employees to better coordinate their work, avoid duplication of labor and reduce the consumption and waste of material resources. Networked and digital sales platform can also improve the efficiency of enterprise sales and after-sales service, and reduce waste of material resources. At the same time, the online sales platform enables enterprises to process orders in batches and optimize distribution routes and methods when delivering goods, thereby reducing carbon emissions and fuel consumption.

Second, ICT investment can help enterprises improve their ability to innovate green technology and thus promote their green development. Enterprises can use the internal production equipment network to strengthen the monitoring of production equipment and find potential problems in production and operation, and feedback this information to the internal departments of the enterprise, so as to facilitate the innovation of production process and production technology. The internal department network of enterprises can enhance the ability of enterprises to coordinate and integrate innovative resources such as human, material and information, thus promoting green innovation of enterprises. In addition, companies can use external network platforms, such as collaborative innovation platforms, to work with other companies in their supply chains or with research institutes to jointly develop advanced green technologies [32].

There are other relevant theories that can be used to explain the intrinsic link between ICT investment and corporate green development. Based on the resource-based view, the resources, capabilities or endowments of enterprises are heterogeneous, and it is the heterogeneous resource endowments that bring higher performance or market competitiveness to enterprises [33]. As a key resource of enterprises, ICT capital plays an important role in improving the efficiency of resource allocation and production, and in promoting green technology innovation. ICT (such as the Internet of Things and data analysis) can optimize resource allocation, improve production efficiency, and reduce the negative impact of production on the environment. The introduction of ICT can also enhance the innovation ability of enterprises, so that they can develop more environmentally friendly products and processes, thereby gaining a competitive advantage in the market. Innovation diffusion theory believes that innovation is an important driving force for economic and social development, and the relative advantages and compatibility of new innovations are important factors affecting the speed of people's adoption of this innovation [34]. First of all, ICT is regarded as an innovation, which can improve the operational efficiency of enterprises and promote the development and application of green technology, which are good for promoting the green transformation of enterprises. Secondly, in promoting the green development of enterprises, ICT investment has obvious advantages in terms of cost-effectiveness, flexibility of solutions and data-driven decision-making compared with investment in pollutant treatment facilities

and replacement of energy-saving equipment [35]. And the results of the green transformation of enterprises through ICT investment are also compatible with the sustainable development goals of enterprises. Therefore, the characteristics of the above ICT investment will accelerate the speed of enterprises' investment in ICT. Finally, from the perspective of the application process of ICT innovation, as more and more enterprises use the advantages of ICT to promote green development, this will form a demonstration effect and encourage other enterprises to follow up ICT investment and achieve green development.

However, ICT products are not completely environmentally friendly in the operation process, and their energy consumption and pollutant emissions are high, which may also hinder green development of enterprises [36]. Metcalfe's law holds that the value of the network is proportional to the square of the number of nodes in the network, that is, the value of the network is an exponential function of the number of nodes in the network. The characteristics of the exponential function show that the value of the network will increase rapidly only when the number of nodes in the network reaches a certain value. ICT investment has an important impact on enterprise production, innovation and energy conservation and emission reduction behavior mainly through network effects, such as the enterprise production equipment network, the internal department network of the enterprise, and the network formed by the enterprise and external related entities mentioned above. As more production equipment within the enterprise is interconnected, more internal departments within the enterprise are interconnected, and enterprises are interconnected with more external stakeholders, enterprises can greatly improve production efficiency and green technology innovation level through ICT investment, thus achieving energy conservation, emission reduction and green development. When the level of ICT investment is low, due to the small network effect, the role of ICT investment in promoting the green development of enterprises may also be small. At this time, the high energy consumption characteristics of ICT products still have a negative impact on the environment. In summary, the impact of ICT investment on the green development of enterprises may not be significant. Therefore, this paper puts forward hypothesis 1:

Hypothesis 1: ICT investment can promote green development of enterprises in high-pollution industries, but this promotion has a threshold effect. Only when ICT investment exceeds the threshold value can it promote green development of enterprises.

## 3.2 Analysis of the mechanism of ICT investment affecting green development of enterprises in high-pollution industries

The theory of green development puts forward several methods to improve the level of green development, and they are the development of green processes and technologies, the design of environmentally friendly products with low life cycle costs, and the implementation of sustainable development strategies [37]. The first two are related to green innovation, and the latter is related to resource utilization. Based on the theory of green development, this paper analyzes the mechanism of production efficiency and green technology innovation.

**3.2.1 The mechanism of production efficiency.** On the one hand, ICT products have the advantages of knowledge-intensive and efficient circulation and the use of ICT products can optimize the types of enterprise factor inputs and improve the efficiency of resource allocation and production efficiency. Specifically, first of all, ICT products have a substitution effect on traditional production factors such as non-ICT capital and labor. After the enterprise realizes the network and digital transformation through ICT investment, it will change the production mode, organization mode and business practice of the enterprise, and then improve the efficiency of resource allocation and production efficiency. For example, enterprises will realize

intelligent production, realize flat enterprise management mode and carry out e-commerce. Second, ICT products have a synergistic effect on traditional production factors. Enterprises will generate data elements when using ICT products. And the industrial Internet platform can collect and process data in a unified way and transmit effective information to different subjects of enterprise production and enterprises related to the upstream and downstream of the industrial chain. These will reduce the degree of information asymmetry and help shorten the time spent on the convergence of factors such as capital and labor. For example, the Internet of Things and big data technologies enable enterprises to monitor production progress in real time, rationally allocate organizational resources and flexibly formulate production plans, thereby improving production efficiency [38]. On the other hand, the improvement of production efficiency will in turn promote green development of enterprises. The negative environmental impact is often related to the excessive consumption of resources and uncontrolled emission of pollutants. The improvement of production efficiency can not only improve the economic benefits of enterprises, but also reduce the redundancy of production input and control the excessive emission of pollutants, which will help promote the green development of enterprises [39]. In summary, this paper proposes hypothesis 2:

Hypothesis 2: Production efficiency is one mechanism through which ICT investment impacts on green development of enterprises in highly polluting industries.

**3.2.2 The mechanism of green technology innovation.** On the one hand, ICT products have the advantages of externality and spillover effect, and ICT investment can enhance the ability of enterprises to innovate green technology. Specifically, first of all, ICT products accelerate the spillover of knowledge and improve the ability of enterprises to absorb knowledge. Nowadays, a large amount of patent knowledge, green innovation experience and industry dynamics can be searched or inquired through professional innovation knowledge platforms. And this kind of patent knowledge and innovation experience increases rapidly over time. Second, the network platform formed by ICT investment helps to achieve knowledge sharing and green technology R&D cooperation. Enterprises can use the internal network platform to strengthen internal personnel communication and cooperation, and to improve the efficiency of enterprise green technology innovation. Enterprises can also use the external network platform to strengthen cooperation with other enterprises in its supply chain, universities at home and abroad, and scientific research institutions, and to jointly develop more environmentally friendly products. Finally, enterprises can obtain consumers' consumption and experience data in green products on the online sales platform, analyze consumers' preferences and suggestions, and then improve the success rate of green product development [40]. On the other hand, green technology innovation also helps enterprises to achieve green development. Green technology innovation can help enterprises to improve production technology and develop more environmentally friendly products in the life cycle [41], which are good for reducing corporate pollution emissions. In summary, this paper proposes hypothesis 3:

Hypothesis 3: Green technological innovation is one mechanism through which ICT investment impacts on green development of enterprises in highly polluting industries.

## 4. Research design

## 4.1 Model construction and sample selection

**4.1.1 Benchmark regression model construction.** Based on the network effect theory, the impact of ICT investment on green development of enterprises in high-pollution industries

is likely to have a threshold effect. At the same time, considering that green development level of the previous period of the enterprise may affect green development level of the current period, this paper chooses the dynamic panel threshold model for research [42,43]. The benchmark regression model is as follows:

$$\text{Ln}CEI_{it} = \beta_0 + \beta_1 L.\text{Ln}CEI_{it} + I(\text{Ln}ICT_{it} \leq \gamma)\beta_2\text{Ln}ICT_{it} + I(\text{Ln}ICT_{it} > \gamma)\beta_3\text{Ln}ICT_{it}$$
$$+ \sum\nolimits_K \theta_K X_{ijpt}^K + \delta_i + \lambda_t + \varepsilon_{it} \tag{1}$$

In Model (1), subscript $i$ refers to the enterprise, $t$ refers to the year, $j$ refers to the industry, and $p$ refers to the province. Adding 'L.' before the variable indicates a lag of one period. Ln$CEI$ represents green development level of the enterprise. Ln$ICT$ represents the level of ICT investment of enterprises, which is both an independent variable and a threshold variable. $I(\cdot)$ represents the indicative function, and when the conditions in the brackets are satisfied, it takes 1, and when it is not satisfied, it takes 0. $\gamma$ is the threshold value and $X$ represents a series of measurable control variables. $\delta$ denotes the fixed effect of the enterprise, $\lambda$ denotes the fixed effect of year, and $\varepsilon$ denotes the random disturbance term.

**4.1.2 Mechanism test model construction.** This paper refers to the common practice in economic research, and verifies the existence of the influence mechanism by testing the influence of independent variable on the mechanism variables [44,45]. Because we only want to verify whether ICT investment can affect the mechanism variable, this paper finally chooses the two-way fixed effect model to test. The test model is shown in Model (2). The independent variable in Model (2) occurs before the mechanism variable, so it is relatively exogenous. Therefore, the endogenous problem of the regression is small.

$$MNV_{it} = \beta_0 + \beta_1\text{Ln}ICT_{it} + \sum\nolimits_K \theta_K X_{ijpt}^K + \delta_i + \lambda_t + \varepsilon_{it} \tag{2}$$

$MNV$ in Model (2) represent mechanism variables, and the meaning of variable subscripts and other variables in Model (2) are the same as those in Model (1).

**4.1.3 Multi-period difference-in-difference (DID) model construction.** The model is used for endogenous test. In 2012, 2013 and 2014, the Chinese government announced three batches of "Smart City" pilot list, which aims to improve the level of urban ICT infrastructure investment. Enterprises can use the ICT infrastructure in the "Smart City" to reduce their carbon emission intensity. Therefore, this paper regards this policy as a quasi-natural experiment and discusses the impact of the policy on the carbon emission intensity of enterprises. Model (3) is constructed for the regression. Model (4) is constructed for parallel trend test of DID model.

$$\text{Ln}CEI_{it} = \beta_0 + \beta_1 PilotSC_{ict} + \sum\nolimits_K \theta_K X_{ijpt}^K + \delta_i + \lambda_t + \varepsilon_{it} \tag{3}$$

$$\text{Ln}CEI_{it} = \beta_0 + \sum\nolimits_m \beta_m PilotSC2_{i,c,t+m} + \sum\nolimits_K \theta_K X_{ijpt}^K + \delta_i + \lambda_t + \varepsilon_{it} \tag{4}$$

In Model (3), $PilotSC$ is a dummy variable. If the city $c$ where the enterprise $i$ is located is a "Smart City" in year $t$, $PilotSC$ takes 1, otherwise 0. In Model (4), $PilotSC2$ represents the constructed policy occurrence period dummy variable, and the value of $m$ is all integers from -6 to 6. Earlier or later years are merged into the 6th year before or after the actual policy period. In order to avoid the problem of multicollinearity, $PilotSC2_{i,c,t-1}$ is deleted in the actual regression.

**4.1.4 Sample selection.** Since China's new accounting standards began to be implemented in 2007, and the latest data on the level of green development of enterprises are only updated to 2019, this paper finally uses the sample of listed companies in the high-pollution industries

**Table 1. High pollution industry list.**

| Industry type | Industry name |
|---|---|
| Light Industry (6) | (1) agricultural and sideline food processing industry; (2) food manufacturing industry; (3) textile industry; (4) leather, fur, feather (velvet) and their products industry; (5) papermaking and paper-products industry; (6) nonmental mineral product industry. |
| Heavy Industry (5) | (1) petroleum processing, coking and nuclear fuel processing industry; (2) chemical raw materials and chemical products manufacturing industry; (3) ferrous metal smelting and rolling processing industry; (4) non-ferrous metal smelting and rolling processing industry; (5) electricity and heat production and supply industry. |

of China's Shanghai and Shenzhen A-shares from 2007 to 2019 for research. Enterprises in high-pollution industries are paying more and more attention to the important role of ICT investment in promoting green development. Compared with enterprises in other industries, their carbon emissions are greater, so their potential to use ICT investment to reduce carbon emission intensity will be greater [11]. In 2019, China's high-polluting industries emitted 8.07 billion tons of carbon dioxide, accounting for 82.4% of total industrial emissions. The delineation of high-pollution industries is based on the standards in "The Plan on the First National Census on Pollution Sources" published by China in 2007. Table 1 shows 11 highly polluting industries, including 5 heavy industries and 6 light industries. In terms of sample processing, samples that were ST, ST* and PT during the sample period were deleted. The samples with total enterprise assets, number of employees and ICT investment less than or equal to 0 during the sample period were deleted. Due to the serious lack of data on economic indicators in Tibet, the samples of listed companies in Tibet were excluded.

## 4.2 Variable construction

**4.2.1 Dependent variable.** Considering that carbon neutralization is the main goal of current environmental governance in China and carbon emission intensity can reflect the dual connotation of "green" and "development", this paper finally uses the logarithm of carbon emission intensity (Ln*CEI*) to measure green development level of enterprises [46,47]. Carbon emission intensity (CEI) = carbon emissions of enterprises / operating income of enterprises. The larger the Ln*CEI*, the lower the level of green development of enterprises. Referring to the idea of Chen, this paper uses the carbon emissions at the industry level to calculate the carbon emissions of enterprises [46]. Carbon emissions of enterprises = (carbon emissions of the industry in which the enterprise is located × the operating cost of the enterprise / the main business cost of the industry in which the enterprise is located).

**4.2.2 Independent variable.** The data of enterprise ICT investment in existing literature are mostly obtained through sample surveys, leading to a short time span of the research sample [48]. In order to obtain more data for consecutive years, drawing on the idea of Li et al. and Ye et al., this paper use the logarithm of the sum of the capital stock of the three types of ICT products (Ln*ICT*), namely, computer equipment, communication equipment and software to measure the level of enterprise ICT investment [11,14]. This data is obtained by manually collating the data in the appendix of the financial statements of listed companies in the RESSET database.

**4.2.3 Mechanism variables.**

1. Production efficiency. TFP represents the ability of an enterprise to maximize output without increasing resource input. Therefore, this paper uses the logarithm of enterprise TFP (Ln*TFP*) to measure the production efficiency of and choose the Levinsohn-Petrin (LP) method to estimate TFP, which can well reduce endogeneity and sample loss [49].

2. Green technology innovation level. The number of green invention patents shows the substantive green technology innovation level of enterprises [50]. Because the number of applications for green invention patents has a large fluctuation relative to the number obtained, referring to the ideas of Hoang et al. and Pan et al., this paper uses the number of green invention patents obtained by enterprises plus 1 and takes the logarithm (Ln*GI*) to measure the level of green technology innovation of enterprises [51,52].

**4.2.4 Control variables.**   Drawing on the ideas of Xu et al. and Ye et al., this paper selects other variables that may affect green development of enterprises from the provincial, industry and enterprise levels [9,11]. Firstly, the control variables at the provincial level include provincial economic development level (Ln*Pgdp*), provincial resource endowment (*Rse*) and provincial environmental regulation level (*Evr*). Ln*Pgdp* is measured by the logarithm of the real per capita GDP of the province. *Rse* is measured by (the sum of fixed asset investment in provincial agriculture and mining industry / fixed asset investment in the whole society of the province). *Evr* is measured by (the amount of investment completed in provincial industrial pollution control / provincial GDP × 100).

Secondly, the control variable at the industry level is the degree of industry competition (*HHI*). *HHI* is measured by the industry Herfindahl index, and the larger the value, the smaller the degree of competition in the industry.

Thirdly, the control variables at the enterprise level include energy consumption scale (Ln*TEC*), enterprise size (Ln*Size*), enterprise per capita assets level (Ln*CI*), asset-liability ratio (*Lev*), return on equity (*ROE*), and ownership concentration (*OC*). Ln*TEC* is measured by the logarithm of the total energy consumption of the enterprise. Consistent with the idea of obtaining corporate carbon emissions, the total energy consumption of enterprises is also calculated through the total energy consumption at the industry level. Ln*Size* is measured by the logarithm of the number of employees in the enterprise. Ln*CI* is measured by the logarithm of (corporate total assets / number of employees). *Lev* is measured by (total corporate liabilities / total corporate assets). *ROE* is measured by (corporate net profit / corporate net assets). *OC* is measured by the proportion of the largest shareholder's shareholding. The data sources and expected directions of all variables in this paper are shown in Table 2.

## 4.3 Variable description and model test

**4.3.1 Variable description.**   In order to reduce the influence of extreme values, this paper carries out the Winsor tail reduction of the first 1% and the last 1% of all variables. The results of descriptive statistics of variables are shown in Table 3, where the difference between the maximum and minimum values of all variables and their average values is within a reasonable range. The minimum values of Ln*CEI* and Ln*ICT* are not much different from their average and maximum values, indicating that the carbon emission intensity of enterprises in highly polluting industries is not too low and their ICT investment also has a certain scale.

**4.3.2 Model test.**   Firstly, the multicollinearity test of independent variable and control variables is carried out, and the VIF values of each variable are within 4, indicating that these variables do not have serious multicollinearity problems.

Secondly, the panel threshold effect test is carried out. In view of the fact that more than 50% of the sample size will be lost when the sample data is transformed into balanced panel data, the xtendothresdpd command is selected to estimate the dynamic panel threshold model of unbalanced panel data. At present, this command can only estimate the single threshold effect. Combined with the conclusion of the previous theoretical analysis, the single threshold model is reasonable. When conducting a single threshold existence test, only the lowest point

**Table 2. Names of variables and sources of data.**

| Variable type | Variable name | Data source | Expected direction |
|---|---|---|---|
| Dependent variable | The logarithm of the carbon emission intensity of the enterprise (Ln*CEI*) | CEADs, EPS and RESSET database | / |
| Independent variable | Enterprise ICT investment level (Ln*ICT*) | RESSET database and manual collation | Negative |
| Mechanism variables | Production efficiency (Ln*TFP*) | CSMAR database | Negative |
| | Green technology innovation level (Ln*GI*) | CNRDS database | Negative |
| Provincial level control variables | Provincial economic development level (Ln*Pgdp*) | National Bureau of Statistics of China | Negative |
| | Provincial resource endowment (*Rse*) | EPS database | Negative |
| | Provincial environmental regulation level (*Evr*) | EPS database and National Bureau of Statistics of China | Negative |
| Industry-level control variable | Degree of industry competition (*HHI*) | CSMAR database | Positive |
| Enterprise-level control variables | Energy consumption scale (Ln*TEC*) | EPS and RESSET database | Positive |
| | Enterprise size (Ln*Size*) | CSMAR database | Negative |
| | Enterprise per capita assets level (Ln*CI*) | CSMAR database | Negative |
| | Asset-liability ratio (*Lev*) | CSMAR database | Indeterminacy |
| | Return on equity (*ROE*) | CSMAR database | Negative |
| | Ownership concentration (*OC*) | CSMAR database | Positive |

of the blue curve can be found to indicate the existence of a single threshold effect. Fig 1 shows the threshold effect test results of the benchmark regression model, where the lowest point of the blue line curve is below the green horizontal line, indicating that the model has passed the single threshold effect test. The threshold value of Ln*ICT* is 12.3.

## 5. Analysis of empirical results

### 5.1 Analysis of benchmark regression results

Columns (1)—(3) in Table 4 are the benchmark regression results of adding control variables at the provincial level, industry level and enterprise level in turn. And column (4) is the benchmark regression result that controls both the firm fixed effect and the year fixed effect. The coefficients of below_Ln*ICT* in columns (1)—(4) are all negative but not significant, and the

**Table 3. Result of descriptive statistics of variables.**

| Variable name | Standard deviation | Minimum | Mean | Maximum | Number of observations |
|---|---|---|---|---|---|
| Ln*CEI* | 1.507 | 3.292 | 6.142 | 9.073 | 6403 |
| Ln*ICT* | 2.357 | 8.480 | 14.65 | 20.00 | 4880 |
| Ln*TFP* | 0.521 | 4.088 | 5.119 | 6.503 | 5599 |
| Ln*GI* | 0.472 | 0 | 0.178 | 2.565 | 6404 |
| Ln*Pgdp* | 0.508 | 9.241 | 10.44 | 11.51 | 6404 |
| *Rse* | 0.0370 | 0.0002 | 0.0339 | 0.200 | 6404 |
| *Evr* | 0.0900 | 0.006 | 0.111 | 0.522 | 6404 |
| *HHI* | 0.0766 | 0.0141 | 0.0956 | 0.457 | 6404 |
| Ln*TEC* | 1.687 | -1.735 | 2.228 | 6.616 | 6403 |
| Ln*Size* | 1.182 | 4.956 | 7.737 | 10.68 | 6397 |
| Ln*CI* | 0.853 | 12.59 | 14.37 | 17.04 | 6397 |
| *Lev* | 0.209 | 0.0514 | 0.453 | 0.921 | 6404 |
| *ROE* | 0.161 | -0.988 | 0.0462 | 0.345 | 6381 |
| *OC* | 0.146 | 0.0985 | 0.356 | 0.750 | 5849 |

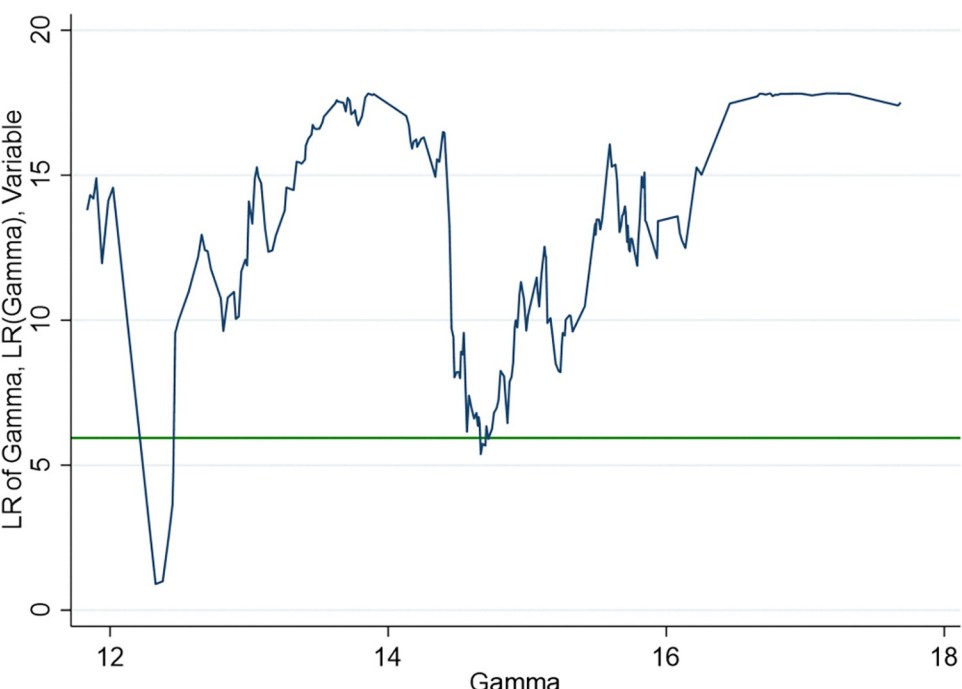

**Fig 1. Estimated value and confidence interval of threshold parameter in benchmark regression model.** Note: The blue line curve represents the likelihood ratio statistic (LR) of different threshold values $\gamma$, and the green horizontal line represents the critical value of LR at the 90% confidence level.

coefficients of above_Ln*ICT* are all negative and significant at least at the 5% significance level. The results show that Ln*ICT* can reduce the carbon emission intensity of enterprises in high pollution industries after exceeding the threshold value, verifying the Hypothesis 1. After ICT investment exceeds the threshold, the network effect becomes prominent, which can ultimately promote green development of enterprises.

In terms of control variables, in columns (4), the coefficients of Ln*Pgdp*, *Rse*, *Evr*, Ln*Size* and Ln*CI* are significantly negative, and the coefficient of Ln*TEC* is significantly positive. This shows that Ln*Pgdp*, *Rse*, *Evr*, Ln*Size* and Ln*CI* can promote the green development of enterprises, while Ln*TEC* will hinder the green development of enterprises. The coefficients of *HHI*, *Lev*, *ROE* and *OC* are not significant, which may be due to their two opposite effects on the green development of enterprises, and ultimately these two effects offset each other.

### 5.2 Analysis of endogeneity and robustness test results

**5.2.1 Endogeneity test.** (1) Using the "Smart City" pilot policy for multi-period difference-in-difference (DID) model regression. The regression results of Column (1) in Table 5 show that the coefficient of *PilotSC* is significantly negative, indicating that the "Smart City" pilot policy can promote the green development of enterprises, which also supports the benchmark regression results of this paper. Fig 2 is the result of the parallel trend test. The results show that only in the second year after the actual policy occurred, the coefficient of *PilotSC2* passed the 5% significance level test and was negative, indicating that the policy has passed the parallel trend test.

(2) Using instrumental variable. In this paper, the topographic relief of the city where the enterprise is located is used as the instrumental variable of the enterprise's ICT investment

**Table 4. Benchmark model regression results.**

| Variables | (1) LnCEI | (2) LnCEI | (3) LnCEI | (4) LnCEI |
|---|---|---|---|---|
| L.LnCEI | 0.0621 | 0.0478 | -0.226** | -0.226** |
|  | (0.68) | (0.53) | (-2.48) | (-2.45) |
| below_LnICT | -0.0218 | -0.0329 | -0.0451 | -0.0436 |
|  | (-0.99) | (-1.38) | (-1.56) | (-1.52) |
| above_LnICT | -0.0346** | -0.0447*** | -0.0515** | -0.0504** |
|  | (-2.36) | (-2.62) | (-2.26) | (-2.23) |
| LnPgdp | -0.855*** | -0.848*** | -0.902*** | -0.908*** |
|  | (-7.22) | (-7.08) | (-4.59) | (-3.22) |
| Rse | 0.154 | -0.323 | -5.982** | -6.018** |
|  | (0.06) | (-0.13) | (-2.46) | (-2.40) |
| Evr | -1.205*** | -1.257*** | -0.999*** | -0.989*** |
|  | (-7.71) | (-7.80) | (-5.84) | (-5.67) |
| HHI |  | -0.638 | 0.427 | 0.451 |
|  |  | (-1.23) | (0.72) | (0.72) |
| LnTEC |  |  | 0.294*** | 0.300*** |
|  |  |  | (5.86) | (4.62) |
| LnSize |  |  | -0.464*** | -0.476*** |
|  |  |  | (-3.45) | (-3.27) |
| LnCI |  |  | -0.356*** | -0.366*** |
|  |  |  | (-2.93) | (-2.76) |
| Lev |  |  | 0.390 | 0.390 |
|  |  |  | (1.35) | (1.36) |
| ROE |  |  | 0.0203 | 0.0216 |
|  |  |  | (0.12) | (0.12) |
| OC |  |  | 1.056 | 1.037 |
|  |  |  | (1.61) | (1.58) |
| Constant | 15.26*** | 15.50*** | 25.52*** | 25.79*** |
|  | (9.37) | (9.64) | (10.75) | (7.22) |
| Enterprise fixed effect | Yes | Yes | Yes | Yes |
| Year fixed effect | No | No | No | Yes |
| N | 4181 | 4181 | 4065 | 4065 |

Note: * means $p < 0.1$,

** means $p < 0.05$,

*** means $p < 0.01$. The $z$ value is in parentheses. Unless otherwise specified, the following is the same.

level, and then two-stage least squares regression (2SLS) is carried out [29]. From the relevant conditions, topographic relief is an important evaluation condition to determine whether a city is suitable for large-scale construction of ICT infrastructure, and the level of ICT infrastructure construction in a city will affect the level of ICT investment of enterprises. From the perspective of exogenous conditions, the carbon emission intensity of enterprises is mainly affected by the location advantage and green technology level of the city where they are located [53]. The location advantage of the city determines the development scale of the advanced manufacturing industry in the city, and the advanced manufacturing industry is often an industry with low carbon emission intensity. As a natural geographical variable, the topographic relief of the city is difficult to directly affect the

**Table 5. The results of endogeneity test.**

| Variables | (1) Multi-period DID regression | (2) The instrumental variable of city topographic relief | (3) Adding *PilotCE* | (4) Adding *PISO14* |
|---|---|---|---|---|
| | LnCEI | LnCEI | LnCEI | LnCEI |
| L.Ln*CEI* | | | -0.224** | -0.235*** |
| | | | (-2.43) | (-2.59) |
| *PilotSC* | -0.0353** | | | |
| | (-2.04) | | | |
| Ln*ICT* | | -0.578** | | |
| | | (-2.09) | | |
| below_Ln*ICT* | | | -0.0392 | -0.0491* |
| | | | (-1.29) | (-1.73) |
| above_Ln*ICT* | | | -0.0472** | -0.0521** |
| | | | (-1.96) | (-2.34) |
| *PilotCE* | | | -0.124 | |
| | | | (-0.71) | |
| *PISO14* | | | | -0.0591 |
| | | | | (-0.71) |
| Constant | 12.42*** | -0.169 | 25.11*** | 25.88*** |
| | (24.03) | (-0.13) | (6.64) | (7.45) |
| Kleibergen-Paap rk LM statistic | | P = 0.0023 | | |
| Cragg-Donald Wald F statistic | | 9.830>8.96(15% maximal IV size) | | |
| Control variables | Yes | Yes | Yes | Yes |
| Enterprise fixed effect | Yes | No | Yes | Yes |
| Year fixed effect | Yes | Yes | Yes | Yes |
| N | 5806 | 4150 | 4065 | 4065 |

Note: The value of *t* is in column brackets of (1) and the value of *z* is in other column brackets.

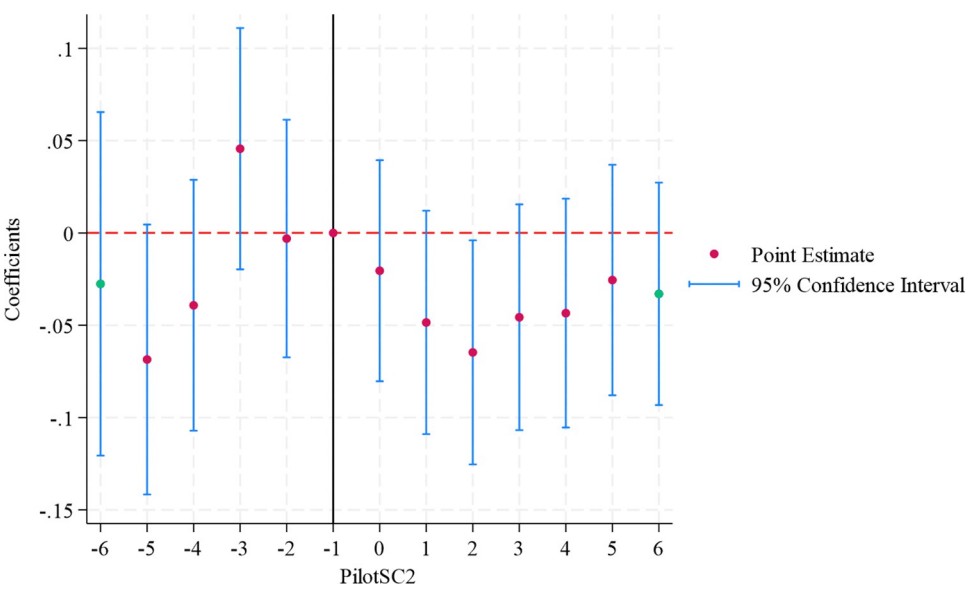

**Fig 2. Parallel trend test of "Smart City" pilot policy.**

carbon emission intensity of enterprises. Columns (2) in Table 5 show that instrumental variable passed the underidentification test (Kleibergen-Paap rk LM statistic) and the weak identification test (Cragg-Donald Wald F statistic). And the results show that the coefficient of Ln*ICT* is significantly negative, indicating that the benchmark regression results are robust.

(3) Adding variables that may be missed. Enterprises may be able to reduce carbon emission intensity after becoming pilot enterprises of carbon emission trading policy, so this paper sets the dummy variable *PilotCE* whether it is a pilot enterprise of carbon emission trading policy and adds it to the benchmark model to regress again. In addition, the carbon emission intensity of enterprises with environmental management system certification (ISO14001) may be lower, so this paper also sets the dummy variable *PISO14* of whether the enterprise has ISO14001 certification to join the benchmark model and regress again. The results of columns (3) and (4) in Table 5 show that the coefficients of above_Ln*ICT* are significantly negative, which further verifies the benchmark regression results. Although the coefficient of below_Ln*ICT* in column (4) is also significantly negative, its absolute value is smaller than that of above_Ln*ICT* and its significance level is relatively lower, which is also basically in line with the conclusion of theoretical analysis.

### 5.2.2 Robustness test.

(1) Replace the dependent variable. The carbon emission intensity of enterprises in this paper only focuses on the carbon dioxide in the pollutant emissions of enterprises, and pays less attention to other pollutant emissions and waste treatment of enterprises. Therefore, this paper uses whether the enterprise is a key pollution monitoring enterprise (*IKP*) to re-measure the green development level of the enterprise. If the green development level of an enterprise is low, it is often a key pollution monitoring enterprise. On the contrary, it is often not a key pollution monitoring enterprise [54]. Because the dependent variable *IKP* is a binary dummy variable, the panel Logit model regression analysis is used. The regression results are shown in column (1) of Table 6, where the coefficient of Ln*ICT* is significantly negative.

(2) Replace the independent variable. This paper uses two methods to re-measure the level of corporate ICT investment. The first method is to use the degree of digital transformation of enterprises, because the degree of digital transformation can also reflect the overall ICT investment level of the enterprise to a certain extent. The degree of digital transformation of enterprises (Ln*WordD*) is measured by adding 1 to the total number of word frequencies related to digital transformation in the annual report of the enterprise and then taking the logarithm. Considering that the word frequency data is count data, the threshold effect is difficult to measure, and this paper uses the two-way fixed effect model to test. The second method is to use the relative level of corporate ICT investment. The absolute level of corporate ICT investment used above may not reflect the scale of corporate ICT investment relative to other types of investment. The relative level of corporate ICT investment (*ICTD*) is equal to the ratio of total corporate ICT investment to total corporate assets. The regression results are shown in columns (2) and (3) of Table 6, where the coefficients of Ln*WordD* and above_*ICTD* are significantly negative, indicating that the baseline regression results are robust.

**Table 6. The results of robustness test.**

| Variables | (1) Replace the dependent variable | (2) Replace the independent variable | (3) Replace the independent variable |
|---|---|---|---|
| | IKP | LnCEI | LnCEI |
| L.LnCEI | | | -0.2594*** |
| | | | (-2.96) |
| LnICT | -0.1216*** | | |
| | (-2.74) | | |
| LnWordD | | -0.0506*** | |
| | | (-7.05) | |
| below_ICTD | | | -0.4098 |
| | | | (-0.46) |
| above_ICTD | | | -0.2763*** |
| | | | (-3.82) |
| Constant | -19.5476*** | 12.08*** | 26.598*** |
| | (-4.15) | (23.54) | (6.63) |
| Control variables | Yes | Yes | Yes |
| Enterprise fixed effect | Yes | Yes | Yes |
| Year fixed effect | Yes | Yes | Yes |
| N | 4267 | 5806 | 4065 |

Note: The value of $z$ is in the parentheses of columns (1) and (3), and the value of $t$ is in parentheses of column (2).

## 5.3 Analysis of mechanism test results

**5.3.1 Mechanism of production efficiency.** Columns (1) and (2) in Table 7 show the regression results with Ln*TFP* as the dependent variable, where Column (1) does not control the year fixed effect. The results show that the coefficients of Ln*ICT* are significantly positive, indicating that ICT investment can improve the production efficiency of enterprises in high-pollution industries. At the same time, improving production efficiency can reduce resource consumption intensity, control excessive emissions of pollutants, and promote green development of enterprises [39]. In summary, production efficiency is indeed one of the influencing mechanisms, which verifies Hypothesis 2.

**5.3.2 Mechanism of green technology innovation.** Column (3) and (4) in Table 7 shows the regression results of Ln*GI* as a dependent variable, where Column (3) does not control the

**Table 7. The results of mechanism test.**

| Variables | (1) LnTFP | (2) LnTFP | (3) LnGI | (4) LnGI |
|---|---|---|---|---|
| LnICT | 0.004*** | 0.003*** | 0.0133*** | 0.0125*** |
| | (3.87) | (3.08) | (3.15) | (2.94) |
| Constant | -0.379*** | 1.680*** | -4.118*** | -2.284*** |
| | (-3.79) | (9.53) | (-11.12) | (-3.39) |
| Control variables | Yes | Yes | Yes | Yes |
| Enterprise fixed effect | Yes | Yes | Yes | Yes |
| Year fixed effect | No | Yes | No | Yes |
| N | 4327 | 4327 | 4454 | 4454 |

Note: The $t$ value is in parentheses.

year fixed effect. The results show that the coefficients of Ln*ICT* are significantly positive, indicating that ICT investment can improve the level of green technology innovation of enterprises in high-pollution industries. At the same time, green technology innovation can promote green development of enterprises by improving production processes and technologies and designing more environmentally friendly products [41]. In summary, green technology innovation is indeed one of the influencing mechanisms, which verifies Hypothesis 3.

## 5.4 Analysis of heterogeneity test results

**5.4.1 Different management levels.** Referring to the idea of Qiu & Yu, the management level is obtained by regression of enterprise management cost to enterprise scale, enterprise export volume and enterprise price markup [55]. The absolute value of the residual of the regression is used to measure the management level of the enterprise, and the smaller the absolute value of the residual, the higher the management level. According to the median value of the sample management level, the high pollution industry enterprises are divided into enterprises with lower management level and enterprises with higher management level. The management level will affect the application efficiency of ICT products. Enterprises with low management level are difficult to integrate ICT products into development strategies and achieve green development [56]. However, enterprises with higher management level can organize employees to learn cutting-edge technology products and unified service standards in time, and promote cutting-edge technology products to play a significant role [57].

Columns (1) and (2) in Table 8 show the regression results of enterprises with different management levels. In the enterprise group with lower management level, the coefficients of below_Ln*ICT* and above_Ln*ICT* are not significant. In the enterprise group with higher management level, the coefficient of above_Ln*ICT* is significantly negative. To sum up, ICT investment has a significant impact on green development of enterprises with higher management level and has no significant impact on the green development of enterprises with lower management level.

**Table 8. The results of heterogeneity test.**

| | (1) | (2) | (3) | (4) | (5) | (6) | (7) | (8) |
|---|---|---|---|---|---|---|---|---|
| | Enterprises with lower management level | Enterprises with higher management level | Light industrial enterprises | heavy industrial enterprises | Non-state-owned enterprises | State-owned enterprises | Traditional ICT investment | Digital technology investment |
| **Variables** | **Ln*CEI*** | **Ln*CEI*** | **Ln*CEI*** | **Ln*CEI*** | **Ln*CEI*** | **Ln*CEI*** | **Ln*CEI*** | **Ln*CEI*** |
| L.Ln*CEI* | -0.306*** | -0.0277 | 0.0175 | -0.286*** | -0.246*** | 0.0588 | -0.0762 | -0.00293 |
| | (-2.70) | (-0.21) | (0.17) | (-2.77) | (-3.34) | (0.46) | (-0.28) | (-0.02) |
| below_Ln*ICT* | -0.0328 | -0.0201 | -0.00694 | -0.0537 | -0.0840*** | -0.0107 | -0.106 | -0.124* |
| | (-1.24) | (-0.94) | (-0.19) | (-1.38) | (-2.82) | (-0.32) | (-1.63) | (-1.89) |
| above_Ln*ICT* | -0.0296 | -0.0335** | -0.0189 | -0.0533* | -0.0792*** | -0.0104 | -0.0981* | -0.107* |
| | (-1.24) | (-2.39) | (-0.61) | (-1.67) | (-3.46) | (-0.38) | (-1.95) | (-1.79) |
| Constant | 23.21*** | 21.10*** | 16.31*** | 16.03*** | 27.40*** | 20.43*** | 18.40** | 31.36*** |
| | (3.17) | (2.79) | (3.08) | (3.34) | (4.94) | (5.16) | (2.50) | (4.41) |
| Control variables | Yes | Yes | Yes | Yes | Yes | Yes | Yes | Yes |
| Enterprise fixed effect | Yes | Yes | Yes | Yes | Yes | Yes | Yes | Yes |
| Year fixed effect | Yes | Yes | Yes | Yes | Yes | Yes | No | No |
| N | 952 | 1042 | 1548 | 2517 | 2190 | 1875 | 2338 | 1727 |

**5.4.2 Different industry characteristics.** According to the different production and technical characteristics of the industry, enterprises in high-pollution industries can be divided into light industrial enterprises and heavy industrial enterprises. Industry characteristics will affect the purpose of enterprise ICT investment. Light industrial enterprises are mostly labor-intensive enterprises, and they are more committed to using ICT products to integrate supply chains, which has a relatively small impact on energy conservation and emission reduction [58]. And they mainly reduce carbon emissions by introducing new production equipment or pollutant treatment equipment, which eventually leads to the insignificant impact of ICT investment on green development of enterprises. Heavy industrial enterprises are mostly resource-intensive enterprises and have a high demand for advanced technology. Thus, they emphasize the use of ICT products to achieve scale manufacturing and agile manufacturing [11], which will profoundly affect the energy conservation and emission reduction behavior of enterprises.

Columns (3) and (4) in Table 8 show the regression results of enterprises in different industries. In the light industrial enterprise group, the coefficients of below_Ln*ICT* and above_-Ln*ICT* are not significant. In the heavy industry enterprise group, the coefficient of above_Ln*ICT* is significantly negative. To summarize, ICT investment has a significant impact on green development of heavy industrial enterprises, and has no significant impact on the green development of light industrial enterprises.

**5.4.3 Different ownership nature.** According to the nature of ownership, enterprises in high-polluting industries are divided into non-state-owned enterprises and state-owned enterprises. The difference in the nature of ownership will also affect the purpose of ICT investment. ICT products can better achieve enterprise carbon emission reduction under the premise of low cost. Non-state-owned enterprises have a high degree of marketization, and they often allocate resources according to the principle of profit maximization [59]. Therefore, the purpose of ICT investment of non-state-owned enterprises includes promoting green development of enterprises. The degree of marketization of state-owned enterprises is relatively low and their resources are not necessarily allocated according to the principle of profit maximization [60]. They do not necessarily take the most cost-effective way, but to achieve green transformation by introducing new production equipment and pollutant treatment equipment. And their main purpose of ICT investment may be to improve economic benefits. This leads to the fact that there is not necessarily a causal relationship between ICT investment and carbon emission intensity of state-owned enterprises.

Columns (5) and (6) in Table 8 show the regression results of enterprises of different ownership types. In the non-state-owned enterprise group, the coefficients of below_Ln*ICT* and above_Ln*ICT* are significantly negative. In the state-owned enterprise group, the coefficients of below_Ln*ICT* and above_Ln*ICT* are not significant. In summary, ICT investment has a significant impact on green development of non-state-owned enterprises, and has no significant impact on the green development of state-owned enterprises. The reason why the coefficient of below_Ln*ICT* in column (5) is also significant may be that ICT investment can increase the degree of asset-lightening of non-state-owned enterprises compared with state-owned enterprises, and the asset-light model is conducive to green technology innovation and green development.

**5.4.4 Different ICT investment types.** 2016 is the year in which digital technology began to play a major role in China's economy and society compared with traditional ICT. Compared with traditional ICT, digital technology has stronger substitutability and synergy to traditional production factors, and its impact on green development of enterprises is greater in scope and intensity [61]. Therefore, this paper divides the samples into two groups according

to the year before 2016 and after 2016 to test the impact of traditional ICT investment and digital technology investment.

Columns (7) and (8) in Table 8 show the regression results of different ICT investment types. Since the span of each time period after segmentation is short, neither of these two regressions controls the year fixed effect. The results show that, in the traditional ICT investment group, the coefficient of below_Ln*ICT* is not significant and the coefficient of above_-Ln*ICT* is significantly negative. In the digital technology investment group, the coefficients of below_Ln*ICT* and above_Ln*ICT* are significantly negative. In conclusion, digital technology investment has a greater impact on green development of enterprises in high-pollution industries than traditional ICT investment, because it does not have to cross the threshold to promote green development of enterprises. In Column (8), the absolute value of the coefficient of below_ln*ICT* is larger than that of above_ln*ICT*. The possible explanation is that the network effect of digital technology investment is greater and the energy consumption of ICT products is still high at this time. When the level of ICT investment is low, the carbon emission intensity of enterprises will be relatively low.

## 6. Conclusion and policy recommendations

### 6.1 Research conclusions

ICT products are changing with each passing day and their role in promoting green development of enterprises is becoming stronger and stronger. However, little literature has studied the ICT investment on green development at the enterprise level. Based on this, this paper first combines the network effect theory and green development theory to explain the impact of ICT investment on green development of enterprises in high-pollution industries. After that, based on the panel data of A-share high-pollution industry enterprises in China's Shanghai and Shenzhen stock markets from 2007 to 2019, this paper empirically tests the non-linear impact of ICT investment on green development of enterprises. The benchmark regression results find that ICT investment can promote green development of enterprises in high-pollution industries, but this promotion has a threshold effect. Only when ICT investment exceeds the threshold value can it promote green development. The mechanism test results show that ICT investment mainly promotes green development of enterprises in high-pollution industries by improving production efficiency and promoting green technology innovation. The heterogeneity test results show that ICT investment has a significant impact on green development of enterprises with higher management level, heavy industrial enterprises and non-state-owned enterprises in high-polluting industries, but has no significant impact on green development of enterprises with lower management level, light industrial enterprises and state-owned enterprises. And digital technology investment has a stronger role in promoting green development of enterprises than traditional ICT investment.

### 6.2 Policy recommendations

From the perspective of enterprises in high-pollution industries, first, enterprises should improve the level of ICT investment. Specifically, enterprises can establish a production management system to monitor resource consumption and pollution emissions in the production process, and to improve their production activities based on real-time feedback data. Enterprises can also actively use the network platform or digital platform to cooperate with multiple stakeholders to improve the ability of green technology innovation. Second, enterprises need to set up a special ICT application department, implement training for employees to apply network and digital products, and ultimately improve the efficiency of applying ICT products. Third, heavy industry enterprises should pay more attention to the role of ICT investment in

promoting green development, and they should actively implement digital transformation measures and use energy and industrial Internet to develop and utilize renewable energy. Light industrial enterprises can also achieve intelligent production and strengthen environmental monitoring and data management through ICT investment, thereby achieving green development in a more economical way. Fourth, non-state-owned enterprises can continue to push for green development through ICT investment, while state-owned enterprises should focus on using ICT products to reduce carbon emissions. Fifth, enterprises should learn and apply cutting-edge digital technologies more actively and continuously monitor and evaluate the effects of new technology applications.

From the government's point of view, first, the government can continue to carry out ICT infrastructure investment to provide a good network and digital environment for enterprises in high-pollution industries. The government can also formulate relevant policies and regulations to help enterprises carry out ICT investment, especially digital technology investment, and to promote ICT resource sharing and cooperative innovation among enterprises. Second, the government can establish a policy support system for network and digital skills training to enhance the skills of ICT product management of enterprises. Third, the government can pay attention to guiding and helping heavy industrial enterprises to invest in ICT products, and strengthen the construction of a series of standard systems such as the collection and analysis of production and management data of enterprise. Fourth, the government can continue to strengthen the internal and external governance of state-owned enterprises and improve their marketization level, so as to promote them to achieve green development through ICT investment.

## Supporting information

**S1 Data.**
(XLSX)

**S1 File.**
(TXT)

## Author Contributions

**Conceptualization:** Yingdong Wang.

**Data curation:** Qingbo Li.

**Formal analysis:** Yingdong Wang.

**Funding acquisition:** Yingdong Wang.

**Investigation:** Yingdong Wang.

**Methodology:** Qingbo Li.

**Project administration:** Yingdong Wang.

**Resources:** Yingdong Wang.

**Software:** Qingbo Li.

**Supervision:** Yingdong Wang.

**Writing – original draft:** Qingbo Li.

**Writing – review & editing:** Qingbo Li.

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
