## [Decision Letter · Decision Letter 0]

1 Oct 2024

PONE-D-24-34856Can ICT investment promote green development? New insights from highly polluting listed enterprises in ChinaPLOS ONE

Dear Dr. Li,

Thank you for submitting your manuscript to PLOS ONE. After careful consideration, we feel that it has merit but does not fully meet PLOS ONE’s publication criteria as it currently stands. Therefore, we invite you to submit a revised version of the manuscript that addresses the points raised during the review process.

**ACADEMIC EDITOR:**

Both reviewers appreciate the relevance and structure of the paper but have identified several areas requiring significant improvements before it can be considered for publication. Based on their feedback, I recommend the following:

<ul><li> 

The connection between ICT investment and green development, particularly in the context of highly polluting industries, needs further theoretical and empirical articulation. Reviewers noted a need for clearer explanations regarding hypotheses and endogeneity concerns.<li> 

The integration of network effect theory into the analysis requires more depth. Additionally, addressing competing or complementary theories will strengthen the paper's contribution to the literature.<li> 

Please address these points and the detailed comments from both reviewers in your revised manuscript

We look forward to receiving your revised manuscript.

Kind regards,

Muhammad Ramzan, PhD

Guest Editor

PLOS ONE

3. We are unable to open your Supporting Information file [data and command.zip]. Please kindly revise as necessary and re-upload.

Additional Editor Comments:

Thank you for submitting your manuscript. Both reviewers appreciate the relevance and structure of the paper but have identified several major areas requiring significant improvements before it can be considered for publication. Based on their feedback, I recommend the following:

1. The connection between ICT investment and green development, particularly in the context of highly polluting industries, needs further theoretical and empirical articulation. Reviewers noted a need for clearer explanations regarding hypotheses and endogeneity concerns.

2. The integration of network effect theory into the analysis requires more depth. Additionally, addressing competing or complementary theories will strengthen the paper's contribution to the literature.

Please address these points and the detailed comments from both reviewers in your revised manuscript.

Reviewers' comments:

Reviewer's Responses to Questions

**Comments to the Author**

1. Is the manuscript technically sound, and do the data support the conclusions?

Reviewer #1: Yes

Reviewer #2: Yes

2. Has the statistical analysis been performed appropriately and rigorously? 

Reviewer #1: Yes

Reviewer #2: No

3. Have the authors made all data underlying the findings in their manuscript fully available?

Reviewer #1: Yes

Reviewer #2: No

4. Is the manuscript presented in an intelligible fashion and written in standard English?

Reviewer #1: Yes

Reviewer #2: Yes

5. Review Comments to the Author

Reviewer #1: The paper offers a well-structured analysis of the influence of ICT investment on green development in high-pollution industries, with a particular focus on green technological innovation. The use of a dynamic panel threshold model and heterogeneity analysis adds significant depth and rigor to the research. However, the theoretical framework, motivation, and policy implications require further refinement to unlock the full potential of the study. I have several major suggestions, and I believe that incorporating these improvements will substantially enhance the overall quality and impact of the paper.

1. The background could provide more specific industry-level challenges related to green development in high-pollution sectors. More emphasis on the environmental and regulatory pressures these industries face would strengthen the case for the study.

The problem statement lacks clarity in terms of how ICT investment specifically addresses these environmental challenges compared to other corporate investments. Further exploration of why ICT investment is uniquely impactful would make the problem more compelling.

2. The contributions section could be more clearly articulated. While the paper mentions filling a gap, it does not sufficiently specify how its findings advance theoretical understanding or provide unique empirical evidence.

The practical contributions to managers or policymakers are underdeveloped, limiting the broader relevance of the findings.

3. The theoretical framework is underdeveloped. While network effect theory is mentioned, it is not thoroughly integrated throughout the analysis. The paper could do more to show how the theory connects ICT investment to firm-level green outcomes.

There is limited discussion of competing or complementary theories that might also explain the relationship between ICT and green innovation, such as innovation diffusion theory or resource-based theory.

4. The theoretical framework is underdeveloped. While network effect theory is mentioned, it is not thoroughly integrated throughout the analysis. The paper could do more to show how the theory connects ICT investment to firm-level green outcomes. There is limited discussion of competing or complementary theories that might also explain the relationship between ICT and green innovation, such as innovation diffusion theory or resource-based theory.

5. There is limited explanation of why certain high-pollution industries were selected or how they differ in terms of ICT adoption and green development potential.

The methodological section does not sufficiently discuss potential biases or limitations in the data, such as the use of proxies for ICT investment or green development.

6. There is limited explanation of why certain high-pollution industries were selected or how they differ in terms of ICT adoption and green development potential. The methodological section does not sufficiently discuss potential biases or limitations in the data, such as the use of proxies for ICT investment or green development.

7. The paper presents a well-structured analysis of how ICT investment influences green development in high-pollution industries, focusing on the role of green technological innovation. The dynamic panel threshold model and heterogeneity analysis add depth to the research. However, the theoretical framework, motivation, and policy implications need further development to fully realize the potential of the study.

Reviewer #2: Can ICT investment promote green development? New insights from

highly polluting listed enterprises in China

The paper raises an important issue and is well written. However, to merit publication in PLOSE ONE, it needs significant improvements in certain aspects. Detailed comments follow:

Major Comments:

1. Line 227: How the authors relate the "threshold effect in networks" with their hypothesized " threshold effect in ICT" needs more articulation or clarification.

2. Line 265: Hypothesis 2 needs paraphrasing. E.g., " Production efficiency is one mechanism through which ICT investment impacts on green development of enterprises in highly polluting industries". The same for hypothesis 3; the phrase "mechanism role" should be improved.

3. Line 308: What the source of this potential endogeneity is, and how testing only the influence of the independent variable avoids this endogeneity needs clarification.

4. Line 311: Can the authors find a statistical/econometric means to establish this influence rather than depending merely on common sense?

5. Line 314: What is ICT shows no significant effect on MNV in model (2). Have the authors checked this before they use MNV as a control variable in Model (3), or they think it does not matter?

6. Line 317: The authors need a reference to substantiate this claim.

7. L 370: This acronym comes all of a sudden. The full phrase should be introduced for the first time, with the acronym in the brackets.

8. L377: The claim "The greater LnPgdp, the stronger the environmental awareness and green consumption ability of local citizens" needs to be supported by a reference.

9. L 381: In the sentence "During most of the sample period, the efficiency of resource

development and utilization in most provinces in China has been high, so Rse is good for green

development of local enterprises" how the first phrase of this sentence leads to the second (so...) is not clear. I mean does the first phrase automatically lead the conclusion that "Rse is good for green development"?

10. L 398: Why +1? Are there enterprises with 0 employees?

11. L 398: The sentence "Large-size enterprises have stronger environmental awareness" needs a reference.

12. L 475: It would be better not to mix models with the results section. All models and their explanation may better be dealt with in section 4, and dedicate section 5 to results and discussion. Moreover, these two sections are very lengthy; they need to be shortened significantly. That way, the paper will be easier to follow.

13. L 502: The authors write "... the carbon emission intensity of an enterprise is difficult to affect the topographic condition of the city where it is located." In relation to the validity of an IV, this sentence should read the other way around, i.e., "the topographic condition of the city is difficult to affect the carbon emission intensity of an enterprise". Moreover, the authors need to elaborate on why they think so.

14. L 506: The authors write "the carbon emission intensity of enterprises cannot directly affect the change of the ratio of ICT assets to total assets of enterprises." The cause and effect relationship should be reversed. The validity of the IV should also be better motivated. Just writing "...cannot directly affect..." is not enough to convince readers that your IV is valid.

15. L 562: Have the authors conducted a statistical test to show that the two values (0.0432 and 0.0504) are significantly different?

16. L 573: The same comment as above.

Minor Comments:

1. Line 93: The phrase "cooperation between enterprises and other enterprises" is ambiguous.

2. Line 112 and elsewhere: The word "literatures" should be replaced with the singular "literature".

3. Line 364: This acronym comes all of a sudden. The full phrase should be introduced for the first time, with the acronym in the brackets.

6. PLOS authors have the option to publish the peer review history of their article (what does this mean?). If published, this will include your full peer review and any attached files.

Reviewer #1: No

Reviewer #2: No

---

## [Author Response · Author response to Decision Letter 0]

15 Nov 2024

We thank the editor and the two reviewers for their constructive comments, which played a great role in improving the quality of this paper. We have carefully revised the paper in response to your comments and uploaded the revision notes as a separate file entitled: ‘Response to reviewers’.

---

## [Decision Letter · Decision Letter 1]

10 Dec 2024

Can ICT investment promote green development? New insights from highly polluting listed enterprises in China

PONE-D-24-34856R1

Dear Dr. Li,

We’re pleased to inform you that your manuscript has been judged scientifically suitable for publication and will be formally accepted for publication once it meets all outstanding technical requirements.

Kind regards,

Academic Editor

PLOS ONE

Additional Editor Comments (optional):

Reviewers' comments:

Reviewer's Responses to Questions

**Comments to the Author**

1. If the authors have adequately addressed your comments raised in a previous round of review and you feel that this manuscript is now acceptable for publication, you may indicate that here to bypass the “Comments to the Author” section, enter your conflict of interest statement in the “Confidential to Editor” section, and submit your "Accept" recommendation.

Reviewer #1: All comments have been addressed

Reviewer #2: All comments have been addressed

2. Is the manuscript technically sound, and do the data support the conclusions?

Reviewer #1: Yes

Reviewer #2: Yes

3. Has the statistical analysis been performed appropriately and rigorously? 

Reviewer #1: Yes

Reviewer #2: Yes

4. Have the authors made all data underlying the findings in their manuscript fully available?

Reviewer #1: Yes

Reviewer #2: Yes

5. Is the manuscript presented in an intelligible fashion and written in standard English?

Reviewer #1: Yes

Reviewer #2: Yes

6. Review Comments to the Author

Reviewer #1: The authors have addressed all important comments in revised version of Can ICT investment promote green development? New insights from highly polluting listed enterprises in China

Reviewer #2: (No Response)

7. PLOS authors have the option to publish the peer review history of their article (what does this mean?). If published, this will include your full peer review and any attached files.

Reviewer #1: No

Reviewer #2: No

---

## [Editor Report · Acceptance letter]

16 Dec 2024

PONE-D-24-34856R1 

PLOS ONE

Dear Dr. Li, 

I'm pleased to inform you that your manuscript has been deemed suitable for publication in PLOS ONE. Congratulations! Your manuscript is now being handed over to our production team.

Kind regards, 

on behalf of

Dr. Yang (Jack) Lu 

Academic Editor

PLOS ONE